# Removal of Pb(II) from Water by FeSiB Amorphous Materials

**Xiang-Yun Zhang [1,2,]*** ⬤, **Liang-Liang He [1], Jin-Ying Du [1] and Zi-Zhou Yuan [1]**

[1] State Key Laboratory of Advanced Processing and Recycling of Nonferrous Metals, Lanzhou University of Technology, Lanzhou 730050, China

[2] Wenzhou Engineering Institute of Pump & Valve, Lanzhou University of Technology, Wenzhou 325105, China

[*] Correspondence: zhangxiangyun86@163.com; Tel.: +86-13109322559

**Abstract:** Amorphous materials have shown great potential in removing azo dyes in wastewaters. In this study, the performance of FeSiB amorphous materials, including FeSiB amorphous ribbons (FeSiB$^{AR}$), and FeSiB amorphous powders prepared by argon gas atomization (FeSiB$^{AP}$) and ball-milling (FeSiB$^{BP}$), in removing toxic Pb(II) from aqueous solution was compared with the widely used zero valent iron (ZVI) powders (Fe$^{CP}$). The results showed that the removal efficiency of all the amorphous materials in removing Pb(II) from aqueous solution are much better than Fe$^{CP}$. Pb(II) was removed from aqueous solution by amorphous materials through the combined effect of absorption, (co)precipitation and reduction. Furthermore, FeSiB$^{AP}$ and FeSiB$^{BP}$ have relatively higher removal efficiencies than FeSiB$^{AR}$ due to a high specific surface area. Although the FeSiB$^{BP}$ has the highest removal efficiency up to the first 20 min, the removal process then nearly stopped due to aggregation.

**Keywords:** amorphous alloys; heavy metal; Pb(II) solution

## 1. Introduction

The rapid development of the modern industry brings more and more threat to our environment. For example, a great deal of lead-containing sewage is produced by smelting, manufacturing, and the oil industry. Excessive lead intake may cause neurological, hematological, and immunological disease. Therefore, these lead ions in the sewage must be removed before it is discharged into the environment. In previous decades, ZVI was widely used to remove various heavy metal ions in water, but it can be easily oxidized, which causes the degradation capability to decay rapidly [1]. In order to improve its degradation capability, various methods have been proposed, including nanotechnology [2,3] and bimetallic technology [4,5], but new limitations such as high production costs, physiological toxicity, and aggregation restrict their application.

Recent reports show that Fe-based amorphous materials in ribbon state or powder state can not only degrade azo dyes from the aqueous solution more efficiently than the wildly used Fe$^{CP}$ but also exhibit relatively stable reusability due to their homogeneous microstructure, thermodynamically metastable nature, and the existence of metalloid elements [6–10]. Furthermore, researchers found that ball-milled amorphous powders can further improve the degradation capacity in degrading azo dyes due to the uneven topography and the stored deformation energy induced by ball milling [11–15]. In view of this, Fe-based amorphous materials in various states may also be able to efficiently remove lead ions in the wastewater.

In this paper, the removal capability and mechanism of lead ion (Pb(II)) from aqueous solution by FeSiB amorphous ribbons were investigated and compared with those of the widely used Fe$^{CP}$. Furthermore, considering that a more specific surface area always leads to a higher reaction efficiency, the removal capacity and mechanism of FeSiB amorphous powders prepared by argon gas-atomization and ball-milling were also studied.

## 2. Materials and Methods

The amorphous ribbons and argon gas-atomized amorphous powders with a composition of $Fe_{78}Si_9B_{13}$ (atomic ratio) were provided by Jiangsu Chuangling Technology Co. and Haoxi Nano technology Co. of China, respectively. The $Fe^{CP}$ was provided by Tianjin Chemical Reagent Research Institute. The amorphous ribbons were cut into pieces of 3 mm $\times$ 3 mm $\times$ 0.03 mm. Of these, part of the ribbons were used directly as removal material for Pb(II), and the rest were annealed at 653 K (below the crystallization temperature) for 1 h in an argon atmosphere and then ground by a high-energy ball mill under an argon atmosphere for 4 h at 200 r/min, with a ball-to-powder mass ratio of 10:1. Afterwards, the pulverized fine powders were filtered by mesh screens of 300 meshes per inch and used as removal material for Pb(II).

Pb(II) solutions were prepared by dissolving $Pb(NO_3)_2$ salt in ultra-pure water. The reaction tests were conducted in 1000 mL beakers placed in a temperature-controlled water-bath device and continuously stirred at 200 r/min. All tests were conducted with removal materials of 1 g/L and Pb(II) solutions of 500 mL, with a concentration of 100 mg/L at pH 5. About 5 mL of solution was extracted at fixed time intervals by a syringe and filtered through a 0.2 μm membrane. Then, they were tested by Inductively Coupled Plasma Optical Emission Spectrometer (ICP-OES, ICAP-7000, Thermo Fisher Scientific, Waltham, MA, USA) to determine their concentration. Cycling experiments were carried out one after another with the same ribbons.

Structure and morphology of the materials were investigated by X-ray diffraction (XRD, D/MAX-2400, monochromated Cu-K$\alpha$ radiation) (Rigaku Ltd., Tokyo, Japan) equipped with an Ni filter and a graphite crystal monochromator, and by scanning electron microscopy (SEM, QUANTA FEG 450, FEI Ltd., Hillsborough, Atlanta, GA, USA) equipped with an Energy Dispersive X-ray Spectrometer (EDS). Specific surface areas of materials were recorded on the Brunauer–Emmett–Teller (BET, ASAP2020, Micromeritics Ltd., Atlanta, GA, USA) using the nitrogen method. Surface elemental information of the materials was determined by X-ray photoelectron spectroscopy (XPS, AXIS SUPRA, Shimadzu Ltd., Manchester, UK).

## 3. Results and Discussion

### 3.1. Removal Capacity of the Materials

Figure 1 shows the XRD spectrum of the four kinds of removal materials. Sharp crystalline peaks of a single $\alpha$-Fe phase are observed on the spectrum of $Fe^{CP}$. The spectra of the other materials present a broad hump in the 2θ range of 40–50°, indicating their amorphous structure. Furthermore, there are tiny crystalline peaks superimposed on the broad humps, indicating that a small part of the samples may crystallize during the preparation process of the amorphous materials.

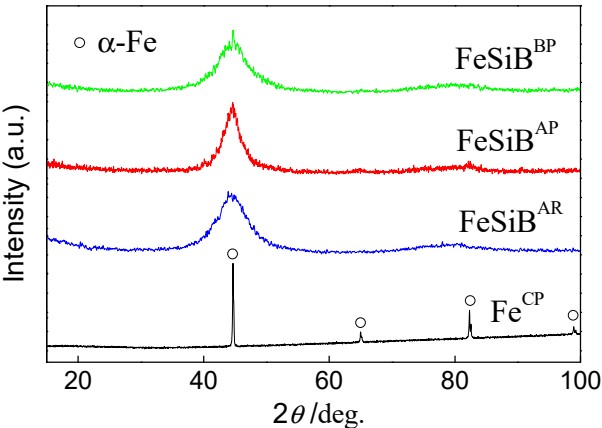

**Figure 1.** XRD spectrum of the removal materials.

Figure 2 shows SEM images of the materials. The morphologies of Fe$^{CP}$, FeSiB$^{AR}$, FeSiB$^{AP}$, and FeSiB$^{BP}$ are shown in Figure 2a–d. All the particles are well-dispersed with no aggregation. The particle size of Fe$^{CP}$ is about 20–50 μm. Grain boundaries can be obviously seen from the enlarged surface morphology in Figure 2a, indicating that these particles are polycrystalline, composed of crystal grains with a size of about 2–5 μm, whereas the thickness of FeSiB$^{AR}$ is about 30 μm (Figure 2b). SEM images of both the top and bottom sides of the ribbon are shown in the inset of Figure 2b. It can be seen that both sides of the ribbon are generally smooth, even if ripples are shown on the roll-contact surface. Compared with the round and ellipsoid morphology of the 10–20 μm sized FeSiB$^{AP}$ particles (Figure 2c), the morphology of the FeSiB$^{BP}$ appears to be rather irregular (Figure 2d). Furthermore, compared with the as-quenched ribbon, thickness of the FeSiB$^{BP}$ reduced to between 5 and 15 μm due to severe mechanical deformation during the ball-milling process, as shown in Figure 2e. Additionally, the surfaces of FeSiB$^{AP}$ are smooth, but the surfaces of FeSiB$^{BP}$ are full of protrusions and microcracks, as shown in the high-magnitude image of Figure 2f.

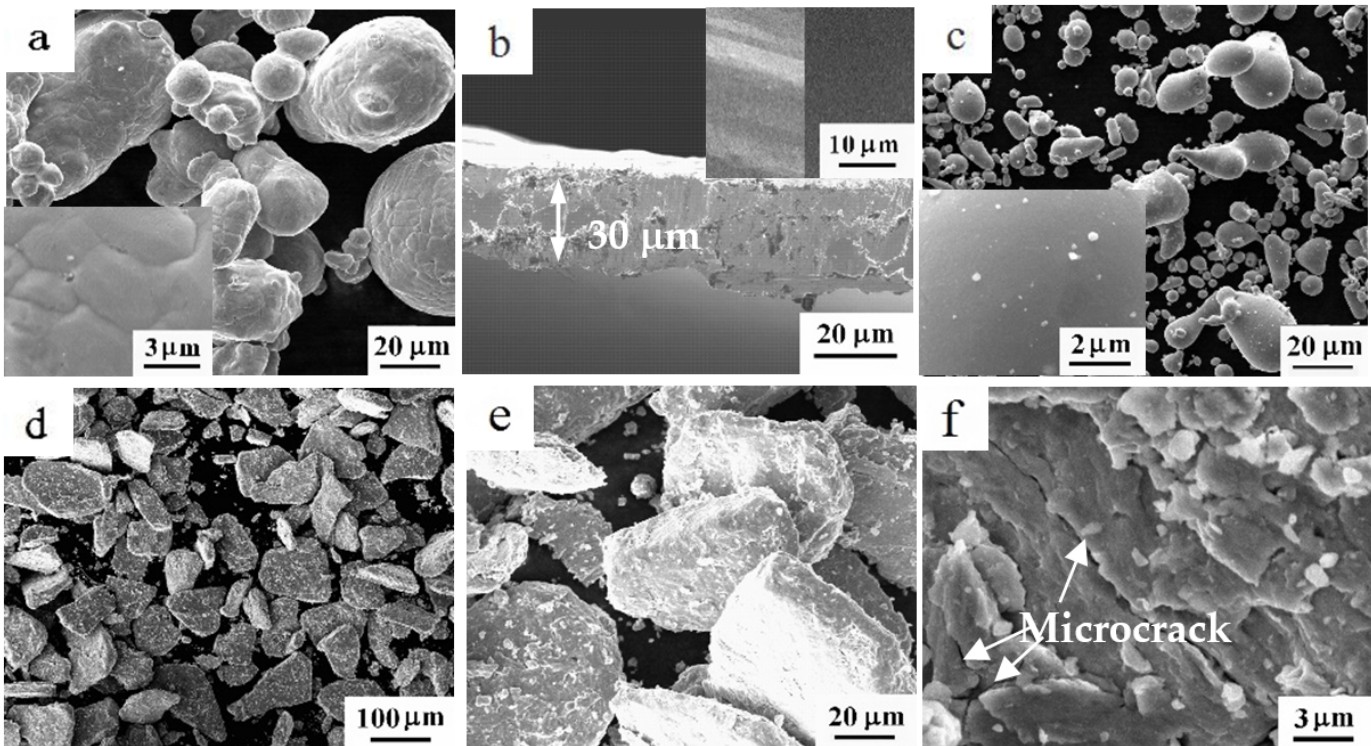

**Figure 2.** SEM images of the removal materials: (**a**) Fe$^{CP}$, (**b**) FeSiB$^{AR}$, (**c**) FeSiB$^{AP}$, and (**d**) FeSiB$^{BP}$. (**e**) and (**f**) are enlarged images of the particles in (**d**). The insets highlight surface details of the materials.

The removal effects of the four materials on Pb(II) solutions at room temperature were investigated. Figure 3 compares the removal performance, dependent on reaction time, for the four materials. $C_t$ is the concentration of Pb(II) at a reaction time of $t$, and $C_0$ is the initial concentration of Pb(II). It can be seen obviously that the removal efficiency of all the amorphous materials are much better than that of Fe$^{CP}$, among which FeSiB$^{BP}$ has the highest removal efficiency up to the first 20 min. Wang et al. [15] also found that the dye degradation capability of Fe-based amorphous ribbons could be enhanced by ball-milling. However, different from Wang's reported results, the removal process seems to stop working and increasing the reaction time cannot effectively reduce the Pb(II) when the Pb(II) concentration is reduced to about 89% at 20 min. Considering the high residual Pb(II) concentration, despite its high removal rate, FeSiB$^{BP}$ is not an optimal removal material

for Pb(II) from aqueous solution. Furthermore, except for FeSiB$^{BP}$, the removal process by other materials can be well-fitted by the first-order reaction equation (Equation (1)) [16]:

$$C_t/C_0 = \exp(-kt) \tag{1}$$

where $k$ denotes the reaction rate constant, which is estimated to be 0.026, 0.079, and 0.116 min$^{-1}$ for the removal processes by Fe$^{CP}$, FeSiB$^{AR}$, and FeSiB$^{AP}$ by nonlinear regression analysis, respectively. Therefore, the removal rates of FeSiB$^{AR}$ and FeSiB$^{AP}$ are approximately 21 and 87 times as fast as the widely used Fe$^{CP}$.

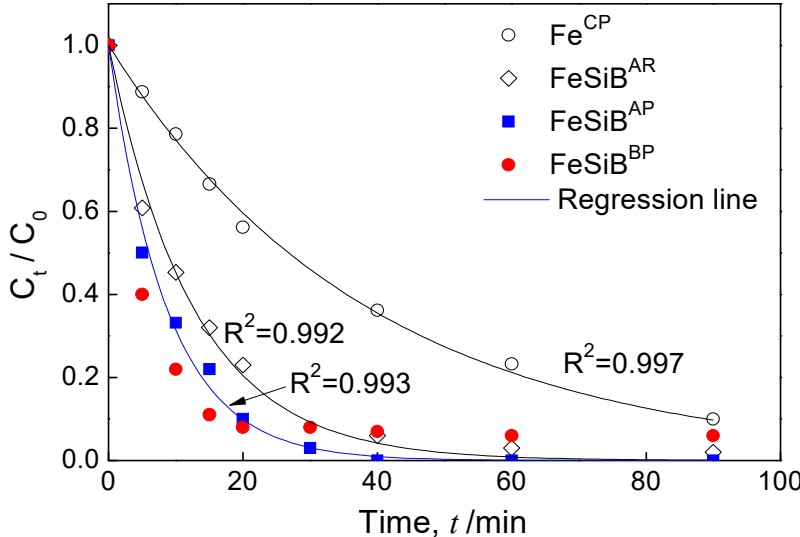

**Figure 3.** The normalized concentration of Pb$^{2+}$ ions dependence on the reaction time.

Considering that the particle size of FeSiB$^{AP}$ is much smaller than that of Fe$^{CP}$, specific surface areas of Fe$^{CP}$ and FeSiB$^{AP}$ are measured to be 0.339 and 0.329 m$^2$/g by a BET analysis, respectively. Thus, the surface area normalized rate constants $k_{SA}$ of Fe$^{CP}$ and FeSiB$^{AP}$ are calculated to be 0.07 and 0.353 L/(m$^2$·min) by the equation: $k_{SA} = k/\rho_a$, where $\rho_a$ is the surface area concentration of the sample [17]. Therefore, the reaction rate of per-unit surface area of FeSiB$^{AP}$ is about 99 times that of Fe$^{CP}$.

### 3.2. Removal Mechanism of the Amorphous Ribbons

Reaction active energy ($\Delta E$) is an effective parameter for exploring the removal mechanism of various reactions. Figure 4 shows the dependence of normalized concentration of Pb(II) on reaction time at different temperatures for Fe$^{CP}$ and FeSiB$^{AR}$. Based on the calculated reaction rate constants at different temperatures, the reaction active energy $\Delta E$ can be derived using the Arrhenius-type equation, $\ln k = -\Delta E/RT + \ln A$, where $R$ is the gas constant, and $A$ is a constant [16]. $\Delta E$ is calculated to be 31.2 and 22.4 kJ/mol for Fe$^{CP}$ and FeSiB$^{AR}$, indicating that the reaction between FeSiB$^{AR}$ and the Pb(II) solution is easier. According to the perspective of thermodynamics, diffusion-controlled reactions in solution have relatively lower activation energies (~8–21 kJ/mol), whereas the surface-controlled chemical reactions have larger activation energies (>29 kJ/mol) [18,19]. Consequently, the reaction between Fe$^{CP}$ and the Pb(II) solution is surface-controlled, whereas interface resistance in the reaction between FeSiB$^{AR}$ and the Pb(II) solution has been greatly reduced. The results are consistent with the report that ZVI removes Pb(II) from solution, mainly by absorption and coprecipitation [20].

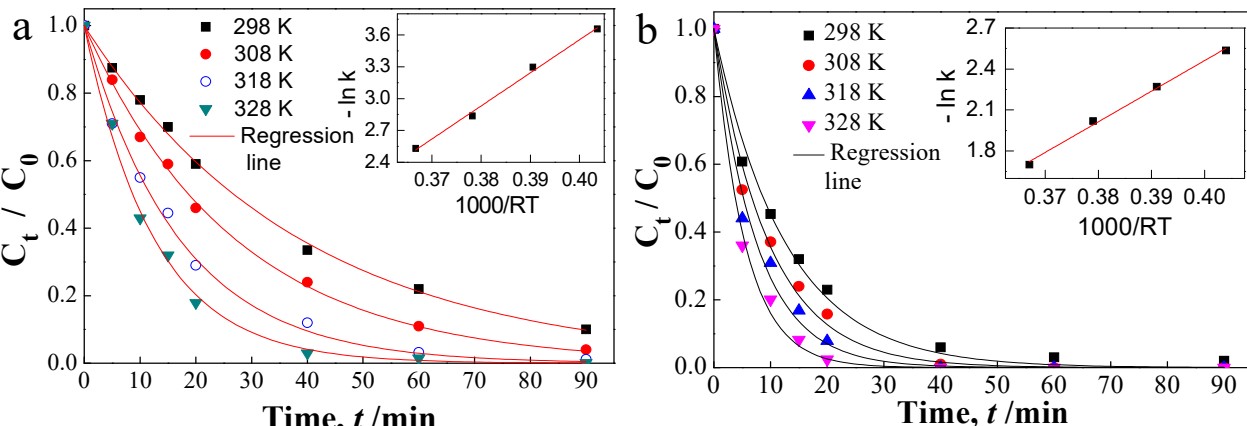

**Figure 4.** Effect of temperature on reaction process of (**a**) Fe$^{CP}$ and (**b**) FeSiB$^{AR}$ in the Pb(II) solution. Insert: Arrhenius plots of $-\ln k$ versus $1000/RT$ for the two kinds of materials, respectively.

In order to investigate the removal mechanism of the amorphous ribbon, XRD patterns of the reacted FeSiB$^{AR}$ are compared with the reacted Fe$^{CP}$. Figure 5 shows XRD patterns of the FeSiB$^{AR}$ and Fe$^{CP}$ after reaction with the aqueous solution for 40 min at room temperature. Diffraction peaks of FeO(OH) appear on both the XRD patterns of the reacted FeSiB$^{AR}$ and Fe$^{CP}$. According to the studies of treatment on Pb(II)-contaminated solution by ZVI, the FeO(OH) complexes with strong flocculation coprecipitate with Pb(II) during the reaction process (Equation (2)) [19]. Therefore, a small part of Pb(II) in aqueous solution was also removed by FeSiB$^{AR}$ by absorption and coprecipitating with Pb(II). Furthermore, compared with Fe$^{CP}$, a large number of diffraction peaks of Pb are found on XRD patterns of the reacted FeSiB$^{AR}$ and the precipitate of FeSiB$^{AR}$, indicating that reduction (Equation (3)) played an important role in the removal process of Pb(II) by FeSiB$^{AR}$. In addition, there are some tiny unknown diffraction peaks found on the XRD patterns of the reacted FeSiB$^{AR}$ and the precipitate. Combined with XRD patterns of the reacted FeSiB$^{AP}$ and FeSiB$^{BP}$, which will be discussed later, these tiny diffraction peaks may represent SiO$_2$ and PbO.

$$\text{Absorption}: \equiv \text{Fe} - \text{OOH} + \text{Pb}^{2+} \rightarrow \equiv \text{Fe} - \text{OOPb} + 2\text{H}^+ \tag{2}$$

$$\text{Reduction}: \text{Fe}^0 + \text{Pb}^{2+} \rightarrow \text{Fe}^{2+} + \text{Pb}^0 \tag{3}$$

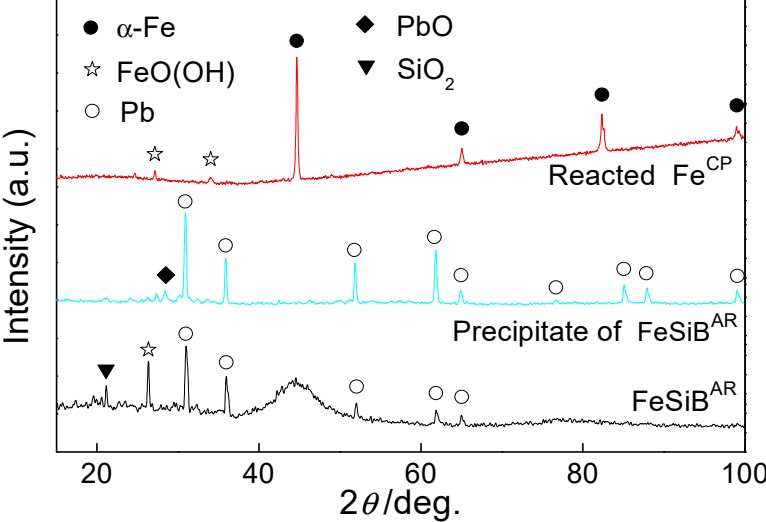

**Figure 5.** XRD patterns of the reacted FeSiB$^{AR}$ and Fe$^{CP}$, and XRD pattern of the precipitate from the aqueous solution reacted with FeSiB$^{AR}$.

In addition, it was reported that hydroxide ions ($OH^-$) were generated by the reaction of $Fe^0$ with $H_2O$ (Equations (4) and (5)) at a pH above 4.5 [21], leading to the increase in the pH of the solution. Then, oxides and hydroxides of iron and lead cover the ZVI particles or form precipitates (Equations (6)–(11)) [2,22,23]. However, some of the reactions cannot be observed by the XRD patterns.

$$Fe^0 + O_2 + 2H_2O - 2e^- \rightarrow Fe^{2+} + 4OH^- + H_2 \uparrow \tag{4}$$

$$Fe^{2+} + O_2 + H_2O + 3e^- \rightarrow Fe^{3+} + O^{2-} + 2OH^- \tag{5}$$

$$Fe^{2+} + 2OH^- \rightarrow Fe(OH)_2 \downarrow \tag{6}$$

$$Fe^{3+} + OH^- \rightarrow Fe(OH)_3 \downarrow \tag{7}$$

$$Fe(OH)_3 \rightarrow FeOOH + H_2O \tag{8}$$

$$FeOOH \rightarrow Fe_2O_3 + H_2O \tag{9}$$

$$(Co)precipitation: Pb^{2+} + OH^- \rightarrow Pb(OH)_2 \downarrow + PbO \cdot xH_2O \tag{10}$$

$$Re-dissolution: Pb(OH)_2 + OH^- \rightarrow Pb(OH)_4^{2-} \tag{11}$$

In view of the limited resolution of XRD, XPS is used to further investigate the surface chemical composition of $FeSiB^{AR}$ and $Fe^{CP}$. Figure 6 shows Pb $4f_{7/2}$ spectra of the reacted $FeSiB^{AR}$ and $Fe^{CP}$. It can be observed that only Pb(II) (138.7 eV) can be found on the surface of the reacted $Fe^{CP}$. However, both Pb(II) (138.5 eV) and Pb(136.7 eV) [24] are found on the surface of the reacted $FeSiB^{AR}$. This discovery further confirms the XRD results showing that the removal of Pb(II) by $Fe^{CP}$ is mainly through absorption and (co)precipitation, while the removal mechanism of Pb(II) by $FeSiB^{AR}$ involves absorption, (co)precipitation, and reduction.

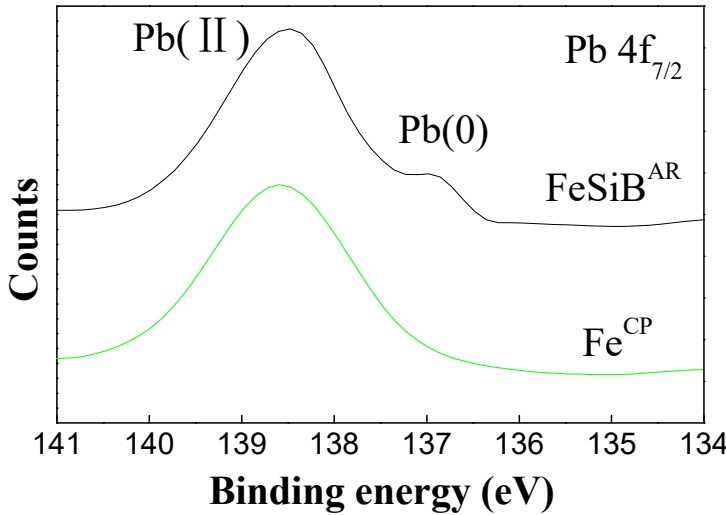

**Figure 6.** Pb $4f_{7/2}$ spectra of the reacted $FeSiB^{AR}$ and $Fe^{CP}$.

Furthermore, the XPS spectra of $FeSiB^{AR}$ before and after reaction are compared in Figure 7. The surface of the as-received $FeSiB^{AR}$ was surrounded by oxide and hydroxide of iron, silicon oxide, and boron oxide. However, the XPS peak of O 1s disappeared after $Ar^+$ sputtered at 0.4 nm/s for 20 s, suggesting that the thickness of the oxidation layer on the surface of the $FeSiB^{AR}$ produced in the atmosphere is about 8 nm. Furthermore, compared with the nominal composition of $Fe_{78}Si_9B_{13}$ amorphous ribbon, the atomic ratios of Fe:Si:B on the surface of the as-received $FeSiB^{AR}$ is 47:37:16, indicating that Si and B atoms are enriched on the surface due to a lower binding energy and the larger diffusion coefficient of Si and B elements [25]. The enrichment of Si and B element may leave an incompact

Fe layer on the surface, which may provide an incompact deposition site for chemical reaction. The possible compositions of the surface layer for the as-received FeSiB$^{AR}$ are ascribed to be Fe$_2$O$_3$(713.45 eV), FeOOH (711.02, 724.35 eV), Fe$^{2+}$ (710.4 eV), Fe$^0$ (706.65 eV), and oxides of boron and silicon [26–28]. After reaction, there are a few changes in the chemical compositions on the surface of the Fe-Si-B$^{AR}$. It contains Fe$_2$O$_3$(713.6, 726.61 eV); FeOOH (711.21, 724.25 eV); Fe$^{2+}$ (710.51 eV); and oxides of boron, silicon, and lead [26–28], indicating the reactions of Equations (4)–(11) involved in the Pb(II) removal process by FeSiB$^{AR}$. Therefore, the combined effect of absorption, (co)precipitation, and reduction lead to the high removal efficiency of FeSiB$^{AR}$, which is similar to the removal mechanism of Pb(II) by nano ZVI [29].

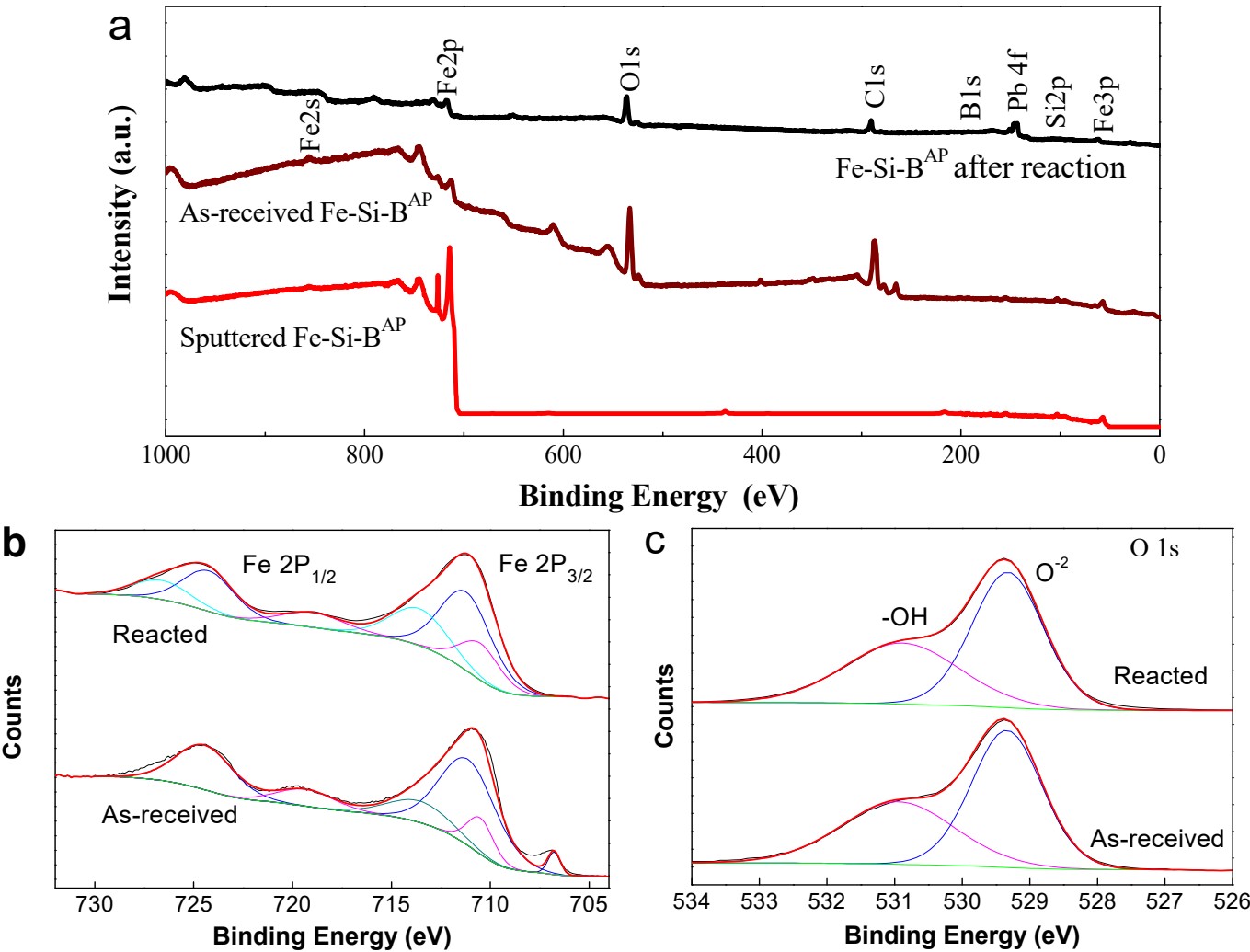

**Figure 7.** XPS spectra of FeSiB$^{AR}$ before and after reaction: (**a**) whole XPS, (**b**) Fe 2p, (**c**) O 1s.

SEM images of the precipitate and reacted FeSiB$^{AR}$ were also observed. Figure 8 shows microscopic morphology of the precipitate. The main features observed are a large number of particles with a size of about 5 μm surrounded by a thin layer of reaction product. It can be seen from the enlarged image in Figure 8b that the particles are surrounded by a villous product layer. EDS elemental analyses in Figure 8d show that the particles are mainly Pb crystallines. This result is in accordance with the XRD analyses that show that the precipitates are mainly Pb. The irregular multi-prism morphology of crystal Pb can be clearly seen from the backscattered electron image of the particles in Figure 8c due to the large difference in atomic number between Pb and the other elements.

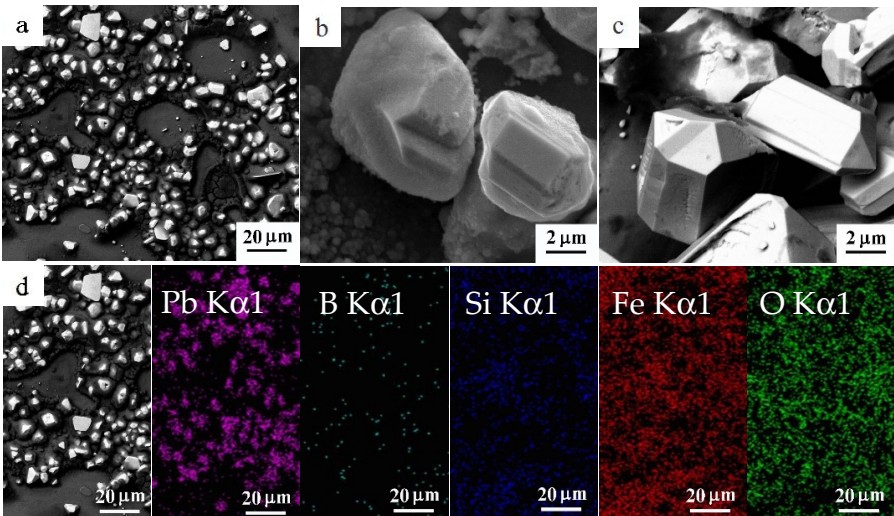

**Figure 8.** (**a**) SEM images of the precipitate after FeSiB$^{AR}$ reacted with the Pb(II) solution. (**b**) Enlarged secondary electron image and (**c**) backscattered electron image of the particles in (**a**). (**d**) EDS elemental mapping images of the precipitate.

Figure 9 shows SEM images of the reacted FeSiB$^{AR}$. Parts of the product layer have peeled off the ribbon (region B) and have formed bright and dark regions on the surface of the FeSiB$^{AR}$, as shown in Figure 9a. The micrograph of region A and B are shown in high resolution in Figure 9b,c, respectively. Cracks have emerged on the surface of the residual product layer, as shown in Figure 9b. Following this, pieces of product layer tear along the cracks due to agitation and expose a fresh amorphous matrix. Then, a new product layer generates, as shown in Figure 9c. It can be seen that lamellar PbO, irregular multi-prism Pb, and worm-like iron oxide or silicon oxide have covered the fresh amorphous matrix. Furthermore, the high-magnitude image of region C in Figure 9d shows that the new product layer is a loose porous structure. The easily detached and loose porous product layer not only helps the elements exchange and accelerates the reaction process but also contributes to the reuse and durability performance of the FeSiB$^{AR}$.

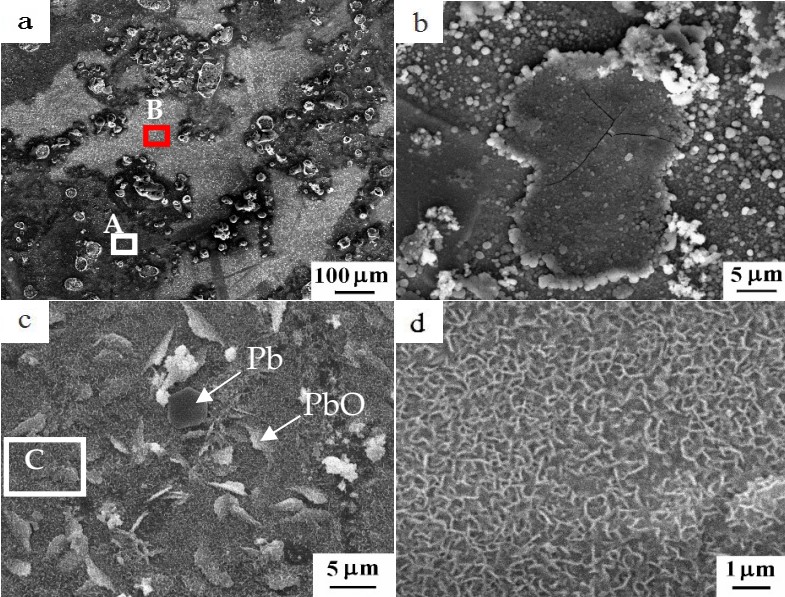

**Figure 9.** (**a**) Surface layer of the reacted FeSiB$^{AR}$ and (**b**,**c**) enlarged views of the corresponding region A and B in (**a**), respectively. (**d**) Enlarged view of the corresponding region C in (**c**).

ZVI is the typical material for water purification, but fast corrosion always leads to rapid decay of its efficiency [28]. To verify the reusability of FeSiB$^{AR}$, repeated removal experiments were carried out for six 90 min cycles. Figure 10 shows the removal efficiency of FeSiB$^{AR}$ for six cycles. Although the removal efficiency gradually decreases after every cycle, removal efficiency up to 54% is still present until the sixth cycle, indicating that FeSiB$^{AR}$ can be reused conveniently for a few times without any treatment.

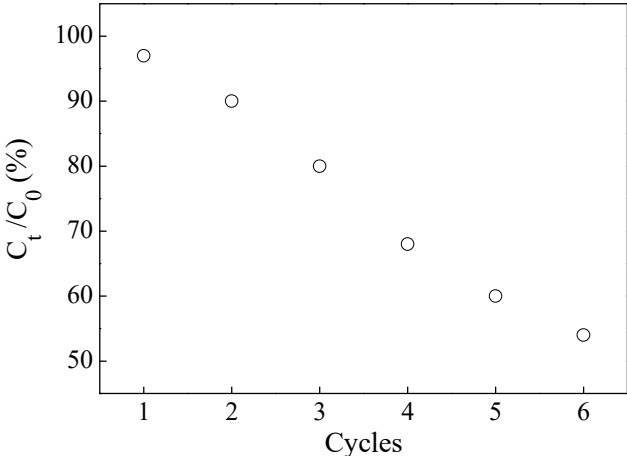

**Figure 10.** Removal efficiency of FeSiB$^{AR}$ reacting with Pb(II) solutions for six 90 min cycles.

The summarized removal mechanism of FeSiB$^{AR}$ is illustrated in Figure 11. FeSiB$^{AR}$ can be rapidly covered by oxidations of Fe, Si, and B in the atmosphere. However, in an oxygenated solution environment, Fe$^{2+}$ and B$^{3+}$ preferentially dissolved into the solution, leaving a porous and incompact layer of Si oxide on the surface of FeSiB$^{AR}$. This not only provides an incompact transport channel for element exchange, but also makes the product layer easy to be peeled off from the ribbon and keeps the ribbon having a relatively stable reusability. Furthermore, the Fe$^0$ beneath the oxide layer acts as an electron donor through the anodic reaction, Fe→Fe$^{2+}$ + 2e$^-$, and releases a steady stream of Fe$^{2+}$ ions into the solution. At the same time, Pb$^{2+}$ is adsorbed on the oxidation layer of FeSiB$^{AR}$ and reduced by Fe$^0$. In addition, the reaction of Fe$^0$ with H$_2$O generates hydroxide ions (OH$^-$) and increases the pH of the solution. Then, oxides and hydroxides of iron and lead may cover the FeSiB$^{AR}$ or form precipitates. In this process, the hydroxides of iron can form surface complexes with Pb(II) and act as an adsorbent of Pb(II), while the precipitate of lead oxide can separate Pb(II) from the solution.

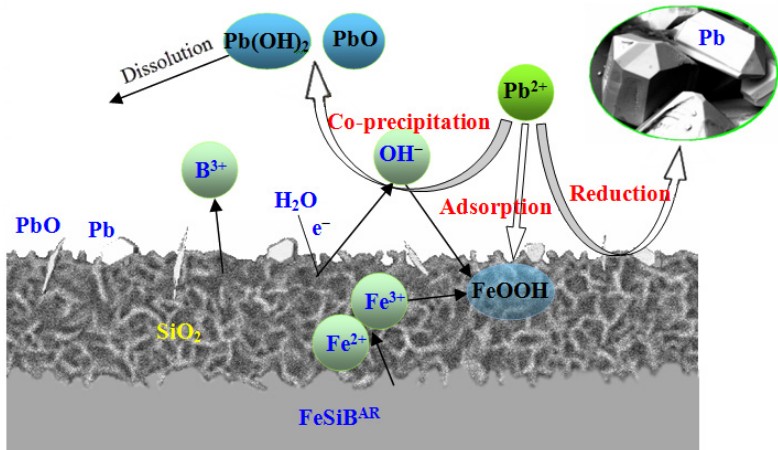

**Figure 11.** Removal mechanism of FeSiB$^{AR}$ in Pb(II) aqueous solution.

In conclusion, the removal of Pb(II) by $Fe^{CP}$ is a surface-controlled process and is mainly controlled by absorption and coprecipitation, but the results suggest that, because of the special amorphous structure, $FeSiB^{AR}$ not only removes Pb(II) in solution by absorption, (co)precipitation, and reduction, but also produces a loose porous product layer. The combination of these effects leads to the high removal efficiency of $FeSiB^{AR}$.

### 3.3. Removal Mechanism of the Amorphous Powders

Figure 12 shows XRD patterns of the $FeSiB^{AP}$ and $FeSiB^{BP}$ after a reaction with the aqueous solution for 40 min at room temperature. The particles are too small to be distinguished from the precipitate, so the samples used for the XRD tests, included the reacted particles and the precipitate. XRD patterns of the $FeSiB^{AP}$ and $FeSiB^{BP}$ show nearly the same diffraction peaks that Pb, PbO, FeO(OH), and $SiO_2$ are superimposing on the broad hump of amorphous particles. The reaction product is consistent with the reaction product of $FeSiB^{AR}$.

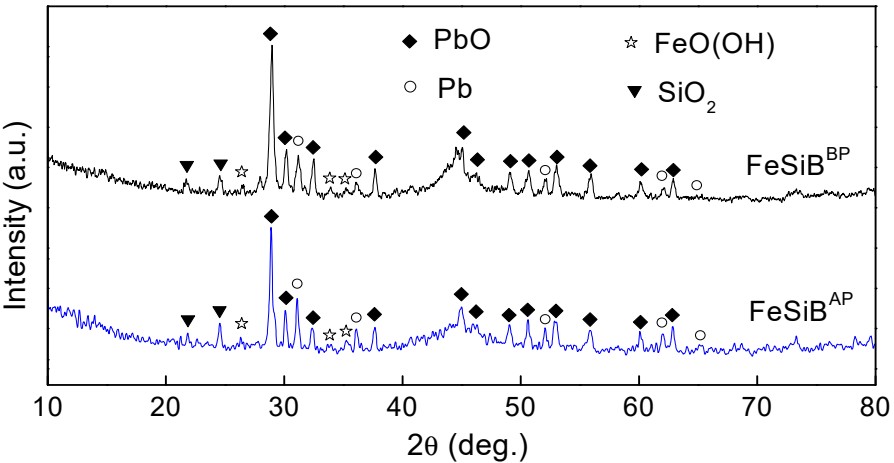

**Figure 12.** XRD patterns of the reacted $FeSiB^{AP}$ and $FeSiB^{BP}$.

Figure 13 shows SEM images of the $FeSiB^{AP}$ after reaction with the aqueous solution for 20 min at room temperature. The reacted $FeSiB^{AP}$ maintains its round or ellipsoid morphology and is covered with heterogeneous phases. It can be seen from Figure 13b that the phases appear to have the same morphologies as the product layers of the reacted $FeSiB^{AR}$. Minor amounts of irregular multi-prism Pb are also present on surface of the reacted $FeSiB^{AP}$, as shown in high resolution in Figure 13c. However, the sizes of the Pb particles are much smaller than those produced by the reaction between $FeSiB^{AR}$ and the Pb(II) solution, which may be related to the short reaction time and relatively dispersed Pb crystalline due to the large specific surface area of the $FeSiB^{AP}$. Additionally, it can be seen from region B that pieces of product layer have peeled off and a new product layer composed of worm-like oxides has generated, as shown in high resolution in Figure 13d. The same phenomenon also appears on the surface of the reacted $FeSiB^{AR}$. Therefore, it can be concluded that $FeSiB^{AR}$ and $FeSiB^{AP}$ have nearly the same reaction mechanism. Compared with $FeSiB^{AR}$, it is the high specific surface area that leads to the higher removal efficiency of $FeSiB^{AP}$.

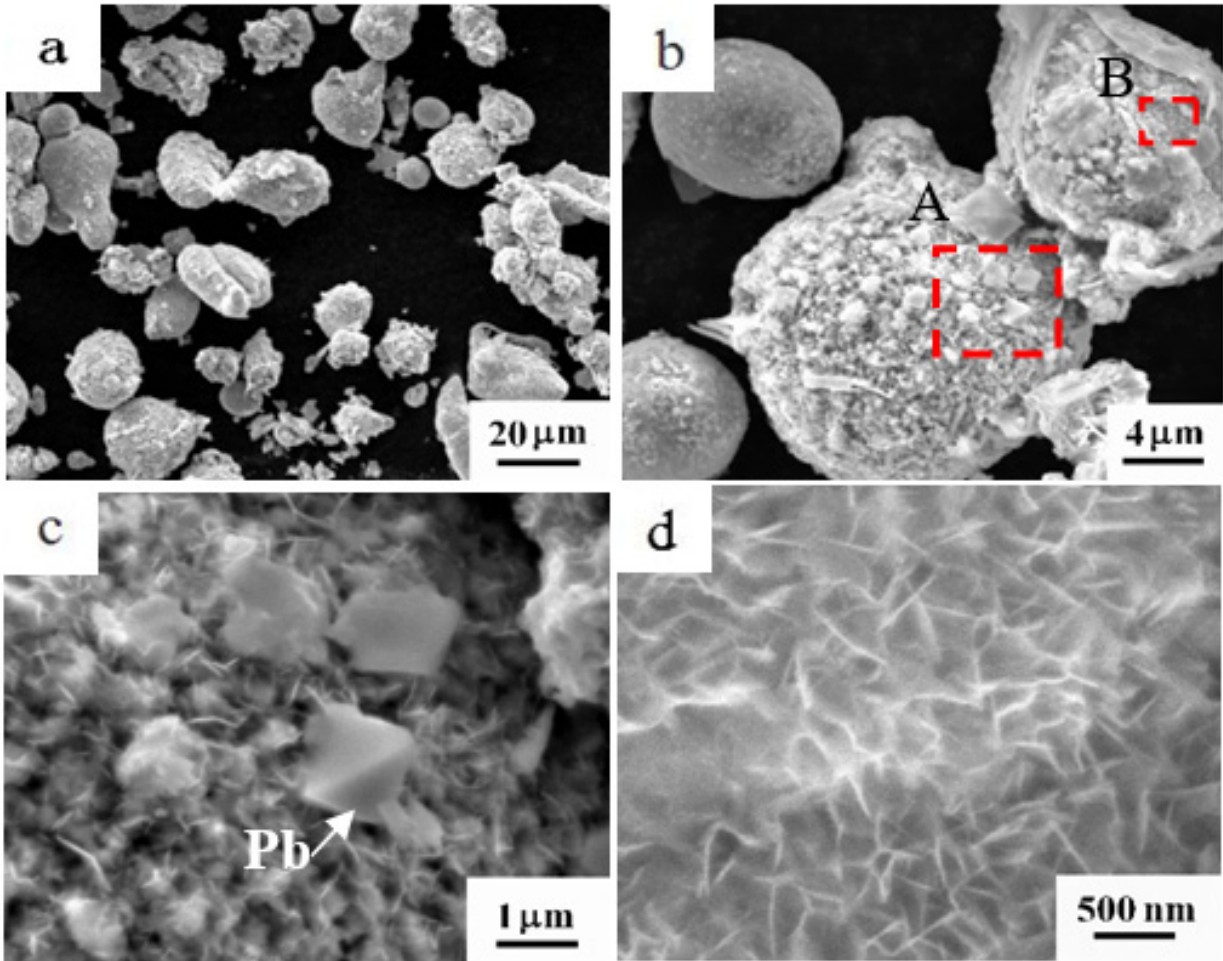

**Figure 13.** (**a**,**b**) SEM images with different magnifications of the reacted FeSiB$^{AP}$ and (**c**,**d**) enlarged views of the corresponding regions A and B, respectively.

It is interesting that the specific surface area of FeSiB$^{BP}$ is measured to be 0.0649 m$^2$/g, corresponding to one fifth of the specific surface area of FeSiB$^{AP}$, but FeSiB$^{BP}$ has the highest removal efficiency after the first 20 min. This phenomenon must be related to the special microstructure induced by the ball-milling process. According to Wang [11], huge residual stress and plastic deformation energy are stored in the FeSiB$^{BP}$ due to intense plastic deformation during the ball-milling process, which may facilitate the reaction activity and contribute to the low reaction active energy. Figure 14 shows SEM images of the FeSiB$^{BP}$ after reaction for 20 min at room temperature. Compared with FeSiB$^{AP}$, the reacted FeSiB$^{BP}$ and the reaction products gather together, as shown in Figure 14a, which may be responsible for the slowing of the removal process after reacting for 20 min.

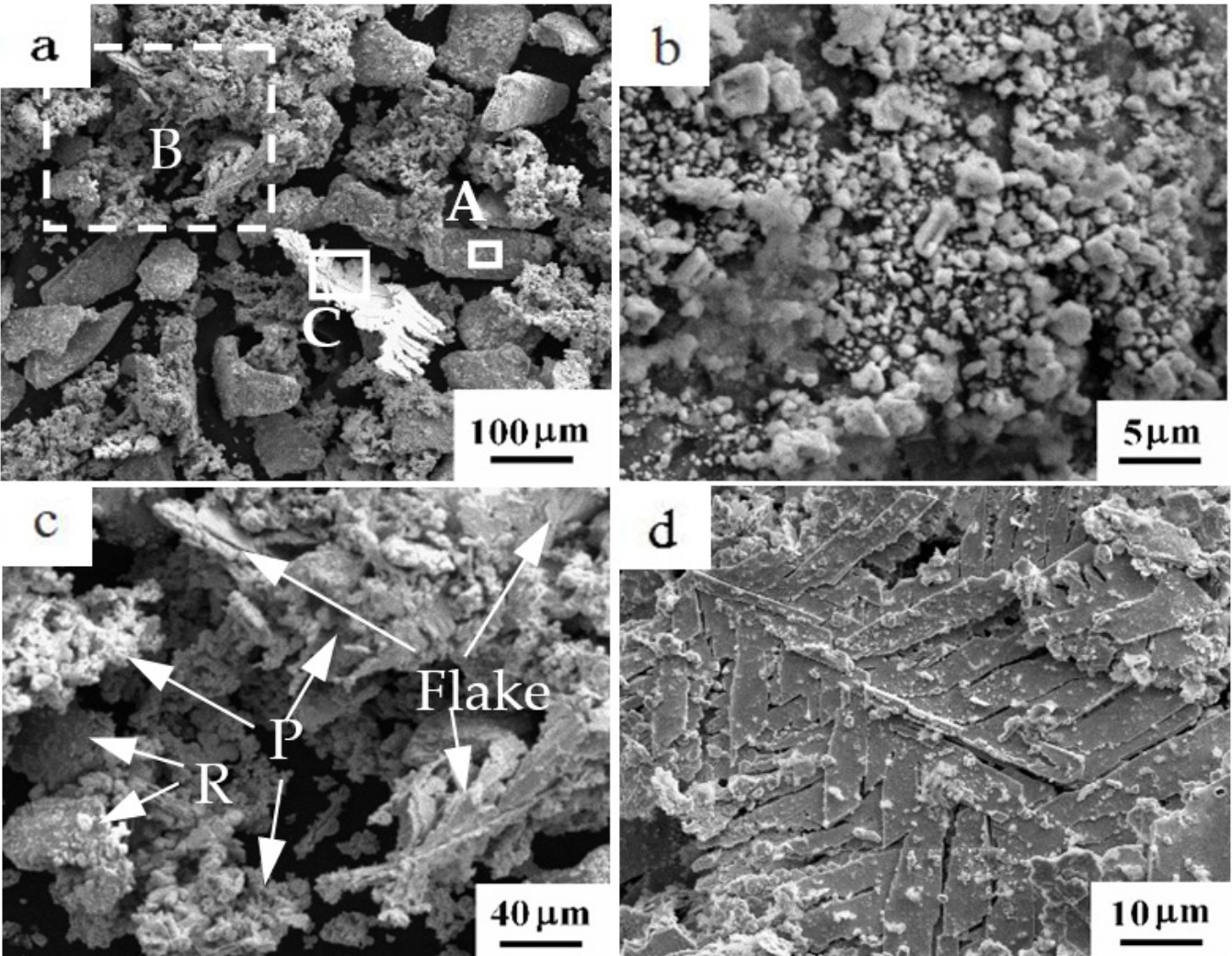

**Figure 14.** (**a**) SEM images with different magnifications of the reacted FeSiB$^{BP}$ and (**b–d**) enlarged views of the corresponding regions A, B, and C, respectively.

It can be seen from the enlarged image of region B in Figure 14c that the aggregation is composed of reaction product particles (marked as P), flakes, and the FeSiB$^{BP}$ residue (marked as R).

The EDS elemental analysis of the reaction product particles in Figure 15 shows that the particles are mainly Pb crystalline, according with the reaction precipitates of FeSiB$^{AR}$. Additionally, there are more Pb particles found on the surface layer of FeSiB$^{BP}$ than on surface of FeSiB$^{AP}$, as shown in Figure 14b. It can be deduced from so large a number of reaction products of Pb crystalline that the reduction is enhanced and plays a main role in the removal process of Pb(II) by FeSiB$^{BP}$, which may be induced by the high deformation energy stored in the FeSiB$^{BP}$.

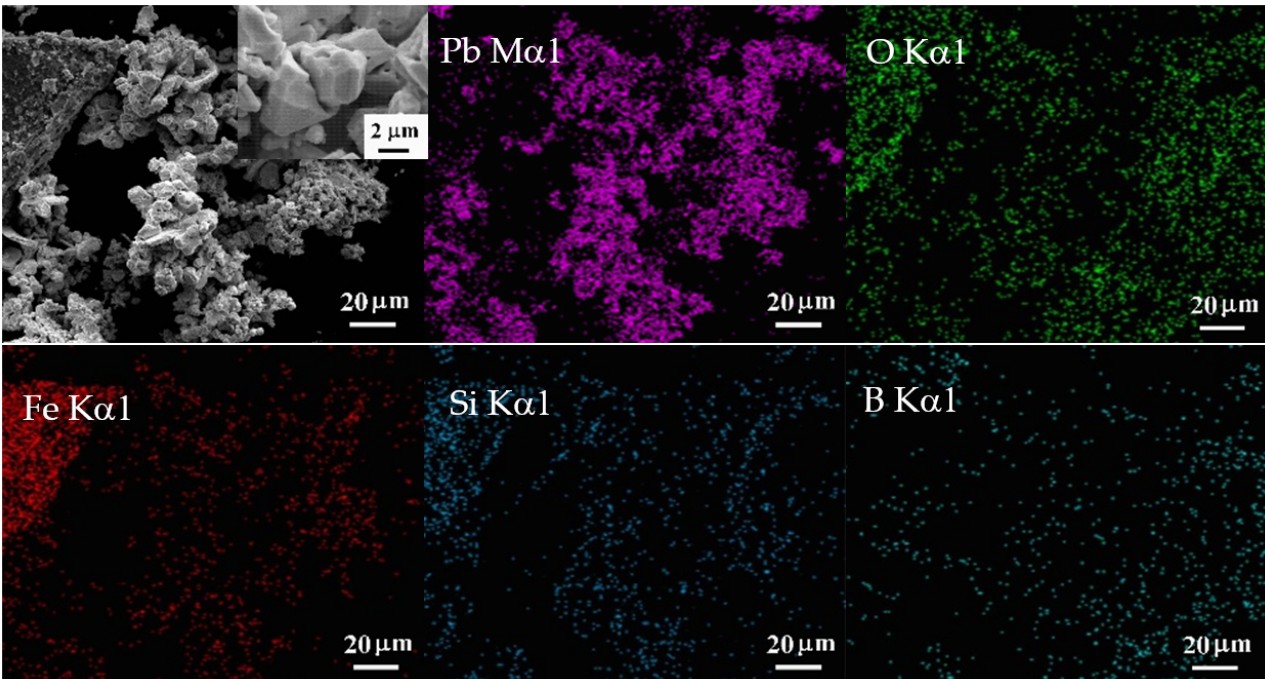

**Figure 15.** SEM image of the reaction products for FeSiB$^{BP}$ and the corresponding EDS elemental mapping images. The insets highlight the details of the particles.

Furthermore, microcracks formed in the ball-milling process on the original FeSiB$^{BP}$ also affect the reaction process. These microcracks may grow and propagate due to the crevice corrosion effect and the huge residual stress in the agitated Pb(II) solution, especially when the FeSiB$^{BP}$ is very thin. The thinner the FeSiB$^{BP}$, the easier it is for the cracks to penetrate the FeSiB$^{BP}$ and break it into small pieces. Figure 14d shows surface details of the flake. It is obvious that the flake would have broken into many small pieces if it had been further agitated. The flakes in Figure 14c have a similar morphology. The increased specific surface areas induced by the breaking of the FeSiB$^{BP}$ may also contribute to the reaction process.

Therefore, even though the original specific surface areas of FeSiB$^{BP}$ are low, the combined effect of high deformation energy stored in the FeSiB$^{BP}$ and increased specific surface areas make them the most reactive material. However, the aggregation of the reaction product and the FeSiB$^{BP}$ residue leads to the rapid decay of its efficiency after reaction for 20 min.

## 4. Conclusions

In summary, the removal efficiency of all the amorphous materials, including FeSiB$^{AR}$, FeSiB$^{AP}$, and FeSiB$^{BP}$, in removing Pb(II) from aqueous solution are much better than the widely used Fe$^{CP}$. Furthermore, it is interesting to find that FeSiB$^{BP}$ has the highest removal efficiency after the first 20 min, but the removal process stops at 20 min, when the Pb(II) concentration is reduced to about 89%.

The removal mechanism of FeSiB$^{AR}$ was discussed, and the results show that, different from the surface-controlled chemical reaction of Fe$^{CP}$, Pb(II) was removed from aqueous solution by FeSiB$^{AR}$ through the combined effect of absorption, (co)precipitation, and reduction. In addition, the product layer on the surface of FeSiB$^{AR}$ is easily detached and is a loose porous structure, which not only accelerates the reaction process but also makes FeSiB$^{AR}$ maintain a relatively stable reusability.

FeSiB$^{AP}$ and FeSiB$^{BP}$ have nearly the same reaction mechanism as FeSiB$^{AR}$, but they have a relatively higher removal efficiency than FeSiB$^{AR}$, due to high specific surface areas. Although FeSiB$^{BP}$ has the highest removal efficiency up to the first 20 min, aggregation of

reaction product and the FeSiB$^{BP}$ residue makes the reaction process nearly halt after the first 20 min of reaction.

This study suggests that all the amorphous materials have great application potential in removal of Pb(II) from wastewater. Despite this, the long-term use of FeSiB$^{BP}$ for Pb(II) removal may need further treatment due to the rapid decay of its efficiency.

**Author Contributions:** Original draft preparation and Funding Acquisition, X.-Y.Z.; supervision, Z.-Z.Y.; investigation, J.-Y.D.; formal analysis, L.-L.H. All authors have read and agreed to the published version of the manuscript.

**Funding:** This work was financially supported by the National Natural Science Foundation of China (NSFC) (No. 52061024 and No. 52161027), the Natural Science Foundation of Zhejiang Province (No. LQ20E010002), the Natural Science Foundation of Gansu Province (No. 21JR7RA260) and the Hongliu Research Funds of Lanzhou University of Technology for Distinguished Young Scholars.

**Institutional Review Board Statement:** Not applicable.

**Informed Consent Statement:** Not applicable.

**Data Availability Statement:** Data sharing not applicable.

**Acknowledgments:** The authors acknowledge all the authors, editors, and reviewers who contributed to this article.

**Conflicts of Interest:** The authors declare no conflict of interest.

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
