# Peer review of "Removal of Pb(II) from Water by FeSiB Amorphous Materials"

_metals, doi:10.3390/met12101740_

Round 1
Reviewer 1 Report
For the future: Fe-based samples cannot be examined using copper radiation due to strong secondary radiation. Use Co, Mo or other radiation, otherwise the information you receive may be incomplete or unreliable
Author Response
I have consulted the experts who specializing in XRD testing about this problem. Decades ago, Fe-based samples must be examined using Co or Mo radiation, but nowadays nearly all the metals can be examined using copper radiation due to the improved configuration of X-ray diffractometer. However, the configuration of the X-ray diffractometer in my article is really not described clearly, so I improved the methods described in my paper.
Reviewer 2 Report
The article entitled “Removal of Pb(â…¡) from water by FeSiB amorphous materials” by Zhang et al. evaluates the performance of FeSiB amorphous materials, including FeSiB amorphous ribbons 10 (FeSiBAR), FeSiB amorphous powders prepared by argon gas atomization (FeSiBAP) and ball-milling 11 (FeSiBBP), in removing toxic Pb(â…¡) from aqueous solution were compared with the widely used zero 12 valent iron (ZVI) powders (FeCP). The write up and presentation of results can be improved as suggested below:
1. Page 1, line 23: Heavy metal ions in the wastewater are harmful to living organisms…
2. Introduction: The introduction section is not informative of the research conducted. The introduction does not provide the background of the research topic
3. Materials and Methods: The section does not detail all methods relevant for the results. No information is presented for methods from which the results were obtained.
4. Results and Discussion: No methods are presented for these results. The methods section should be improved to support these results. The introduction section should be improved to provide a proper background for this research.
5. Summary: The authors miss an opportunity to publish a good paper by laziness in providing background data. For this reason, I do not recommend publication of this article in the current form.
Author Response
Thank you very much for you suggestions. We have realized that our paper does have a lot of points for improvement. And we have carefully revised our “Introduction” and the details of our “Materials and Methods”. The revised sentences are marked with yellow back.

Reviewer 3 Report
The article is well planned and is a logical whole. It touches on a very topical subject, combines modern materials with environmental protection and improvement of the quality of life, which is worth emphasizing. In my opinion, the article may be published after minor changes and answering questions.
Comments:
- how the samples were prepared for XRD and SEM tests (the method of sample preparation may affect the obtained results, the experiment must be clarified so that it can be repeated),
- provide more data about the experiment, perhaps a drawing or a diagram for conducting the experiment and research would be useful,
- is the FeSiBBP sample fully amorphous (in my opinion a clear peak is visible, probably from the aFe phase),
- what databases did the authors use to identify the phases?
Author Response
- The structures of the ribbons and powders before and after removal experiments were directly examined by XRD and SEM without any extra treatment.
- The part of experiment has been refined.
- Yes, under your guidance, I also found this phenomenon. And I have revised the XRD results in the paper.
- The databases used to identify the phases of XRD are the ordinary PDF card from Jade Software.

Round 2
Reviewer 2 Report
Based on the earlier review, the authors answered almost all questions. The paper is almost ready for publication. I still believe the introduction could be expanded to provide more background to the problem statement. The literature coverage of the paper is very thin.
I am also afraid to say that the methods used are not referenced. Referencing methods helps the user of the report to access the original method and to determine if any modifications were done on the method. It helps others to replicate the experiments and validate the data. This is a matter of good scientific principles which must be upheld all the time.
Author Response
Thank you very much for you suggestions. We have expanded the background in the introduction. But the methods used in our paper are ordinary routine experimental process in water treatment, and most of this kind of articles didn't have references. So I wonder if the methods in my article can also be unreferenced.
Reviewer 3 Report
The authors answered my questions. I believe the article may be published in Metals.
Author Response
Thank you very much. ^_^